# The Clinical Impact of Different Types of Preoperative Biliary Intervention on Postoperative Biliary Tract Infection of Patients Undergoing Pancreaticoduodenectomy

**DOI:** 10.3390/jcm13144150

**Published:** 2024-07-16

**Authors:** Min-Jung Wu, Yung-Yuan Chan, Ming-Yang Chen, Yu-Liang Hung, Hao-Wei Kou, Chun-Yi Tsai, Jun-Te Hsu, Ta-Sen Yeh, Tsann-Long Hwang, Yi-Yin Jan, Chi-Huan Wu, Nai-Jen Liu, Shang-Yu Wang, Chun-Nan Yeh

**Affiliations:** 1Division of General Surgery, Department of Surgery, Linkou Chang Gung Memorial Hospital, Taoyuan 333, Taiwan; mpq019@cgmh.org.tw (M.-J.W.); owendisk7541@cgmh.org.tw (Y.-Y.C.); jen.y554@mail.jah.org.tw (M.-Y.C.); brianhung24@cgmh.org.tw (Y.-L.H.); b9602039@cgmh.org.tw (H.-W.K.); m7202@cgmh.org.tw (C.-Y.T.); hsujt2813@cgmh.org.tw (J.-T.H.); tsy471027@cgmh.org.tw (T.-S.Y.); hwangtl@cgmh.org.tw (T.-L.H.); janyy@cgmh.org.tw (Y.-Y.J.); 2Chang Gung University, Taoyuan 333, Taiwan; milk1372@cgmh.org.tw; 3Department of Gastroenterology and Hepatology, Linkou Chang Gung Memorial Hospital, Taoyuan 333, Taiwan; b9002076@cgmh.org.tw

**Keywords:** biliary tract infection, endoscopic retrograde cholangiopancreatography, pancreaticoduodenectomy, percutaneous transhepatic cholangiography and drainage, periampullary tumor

## Abstract

**Simple Summary:**

Post-pancreaticoduodenectomy complications are still a major concern nowadays, and we focused on one of the most catastrophic conditions observed from clinical practice, the postoperative biliary infection, which may cause postoperative intensive care unit admission or mortality. Evaluating the impact on postoperative biliary tract infection from endoscopic retrograde cholangiopancreatography or percutaneous transhepatic cholangiography and drainage before surgery is important to clinical circumstances. Biliary pathogens identified with a positive yield of intra-operative bile culture in patients undergoing pancreaticoduodenectomy were also investigated. The result showed that performing endoscopic retrograde cholangiopancreatography enhances the likelihood of achieving a positive yield from intraoperative biliary culture. Thus, PTCD emerges as a potentially favorable option for patients with obstructive jaundice and is indicated for pancreaticoduodenectomy.

**Abstract:**

**Background**: For patients with obstructive jaundice and who are indicated for pancreaticoduodenectomy (PD) or biliary intervention, either endoscopic retrograde cholangiopancreatography (ERCP) or percutaneous transhepatic cholangiography and drainage (PTCD) may be indicated preoperatively. However, the possibility of procedure-related postoperative biliary tract infection (BTI) should be a concern. We tried to evaluate the impact of ERCP and PTCD on postoperative BTI. **Methods**: Patients diagnosed from June 2013 to March 2022 with periampullary lesions and with PD indicated were enrolled in this cohort. Patients without intraoperative bile culture and non-neoplastic lesions were excluded. Clinical information, including demographic and laboratory data, pathologic diagnosis, results of microbiologic tests, and relevant infectious outcomes, was extracted from medical records for analysis. **Results**: One-hundred-and-sixty-four patients from the cohort (164/689) underwent preoperative biliary intervention, either ERCP (n = 125) or PTCD (n = 39). The positive yield of intraoperative biliary culture was significantly higher in patients who underwent ERCP than in PTCD (90.4% vs. 41.0%, *p* < 0.001). Although there was no significance, a trend of higher postoperative BTI (13.8% vs. 2.7%) and BTI-related septic shock (5 vs. 0, 4.0% vs. 0%) in the ERCP group was noticed. While the risk factors for postoperative BTI have not been confirmed, a trend suggesting a higher incidence of BTI associated with ERCP procedures was observed, with a borderline *p*-value (*p* = 0.05, regarding ERCP biopsy). **Conclusions**: ERCP in patients undergoing PD increases the positive yield of intraoperative biliary culture. PTCD may be the favorable option if preoperative biliary intervention is indicated.

## 1. Introduction

Obstructive jaundice is one of the common presentations in cases of periampullary tumors, owing to the obstruction at the common bile duct secondary to primary cancer or external compression by metastatic lesions [1]. Malignant obstructive jaundice may impose adverse effects, such as impairment of hepatic function, coagulation disturbances, biliary infection, or hepatic failure, on later cancer treatment [2,3]. While pancreaticoduodenectomy (PD) is the treatment with priority, preoperative biliary intervention (PBI) with diagnostic and/or therapeutic purposes may be indicated for selected patients. Biliary intervention, mostly endoscopic retrograde cholangiopancreatography (ERCP) or percutaneous transhepatic cholangiography and drainage (PTCD, has been adopted before pancreaticoduodenectomy (PD) in selected patients. Although PBI has been reported in several studies to carry a higher risk of postoperative morbidities compared to direct surgery [3,4,5,6], it is justified under certain circumstances, such as deep jaundice, cholangitis, and possible differential diagnoses for which surgery may not be the treatment of choice [7,8]. A cohort study by Olecki et al. demonstrated that over 60% of patients undergoing PD had biliary stent placement preoperatively [4]. Therefore, it is a common scenario for surgeons who would accept patients with PBI for surgery and surgeons cannot neglect the impact of PBI.

The most reported negative impact of PBI in both primary studies and meta-analysis was infection-related complications [5,9,10,11,12,13,14,15,16]. There was no obvious statistical significance regarding major complications, such as mortality, reported [4,5,11,12,14]. In addition, there were also studies focusing on the clinical influence of different types of PBI [7,17,18]. Infection-related complications in the aforementioned studies referred to surgical site infection (SSI), which occurs after surgery in the part of the body where the surgery took place (https://www.cdc.gov/surgical-site-infections/about/?CDC_AAref_Val=https://www.cdc.gov/hai/ssi/ssi.html, accessed on 9 April 2024). While enhanced surgical wound care and dressing would improve the superficial SSI, radiological interventions would provide adequate drainage for SSI in cavitary spaces. However, patients cannot receive adequate drainage when encountering post-PD BTI, especially without postoperative external biliary drainage. Without adequate drainage, BTI may be a catastrophic condition that could be accompanied by sepsis or organ dysfunction [19].

Since PBI is currently indicated for selected patients who are candidates for PD, choosing an optimal PBI and preventing patients from relevant complications is imperative. In the present study, we only focused on the most challenging post-PD infection-related complication, post-PD BTI, rather than other SSIs. In our institutional experience, patients who suffered from post-PD BTI usually underwent PBI with ERCP. Therefore, we tried to understand whether scientific evidence would coincide with our clinical observation, namely, that ERCP imposes a higher risk of post-PD BTI than PTCD.

## 2. Materials and Methods

From June 2013 to March 2022, patients diagnosed with periampullary lesions and undergoing PD at Chang Gung Memorial Hospital (CGMH), Linkou Branch, were enrolled in the present study. Patients without complete clinical information or intraoperative bile culture were excluded from the analysis. The Internal Review Board of CGMH approved this tertiary care setting, single-center, retrospective cohort study under reference number 202000980B0. Clinical information, including demographic data (sex, age, indication), American Society of Anesthesiologists Performance Score (ASA-PS), laboratory data, preoperative BTI, information on PBI, details of surgical procedure, pathologic diagnosis, results of microbiologic tests, and relevant outcomes were collected retrospectively from medical records.

### 2.1. Definition of Biliary Tract Infection after Pancreaticoduodenectomy

BTI, or cholangitis, after PD was defined as the association of clinical signs of infection (fever and chills), an elevation of inflammatory serum markers, and abnormal hepatic function tests improving over time under antibiotic therapy [20]. In our study, we focused on BTI within 30 days after surgery and during index hospitalization for PD. The elevation of inflammatory markers and abnormal hepatic function was defined according to the postoperative baseline of individual patients. The laboratory baseline was the first postoperative blood test, which was performed within 48 h after surgery. In addition to clinical and laboratory findings, computed tomography (CT) was arranged for all the patients with suspicious surgical infections. If a cavitary abscess was identified, we did not diagnose patients with post-PD BTI, even with comparable laboratory findings.

### 2.2. Preoperative Biliary Interventions (PBIs) and Surgical Procedures

The preoperative biliary intervention was performed via either ERCP or PTCD for purposes of diagnosis with or without therapy, namely, biliary drainage. Before performing ERCP or PTCD, endoscopists evaluated indications by reviewing existing imaging studies. ERCP is performed via a side-viewing endoscope inserted into the papilla with contrast injected through the ampulla of Vater into the bile ducts. Additional procedures associated with ERCP, including endoscopic papillotomy (EPT), biopsy, endoscopic nasobiliary drainage (ENBD), and endoscopic retrograde biliary drainage (ERBD), were also reviewed and extracted from medical records. PTCD is an ultrasound- or CT-guided technique carried out by interventional radiologists. A fine needle is punctured through the skin into an intrahepatic duct, following the Seldinger technique for drainage placement, and the initial size of the tube is 8Fr if there is no other specific consideration. Patients with cross-over failure between the two treatments were excluded from the study. In our institute, we have not conducted ENBD in the recent decade, and all the patients enrolled in the present study did not undergo this procedure. ERBD in the present work indicated biliary drainage with stent placement. The standard surgical procedure, including classical PD and pylorus-preserving pancreaticoduodenectomy (PPPD), is based on the surgeon’s decision. The operation timing was also determined by surgeons based on each case’s clinical status, primarily 7–21 days after PBI. The variance in timing was contingent upon several factors. Firstly, if complications arose post-biliary drainage, such as pancreatitis, hemorrhaging, or infection, priority was given to stabilizing the patient before proceeding with surgery. Secondly, in cases where patients underwent ERCP biopsy concurrent with drainage, surgery was postponed pending the receipt of biopsy results.

### 2.3. Risk Factor of Post-Pancreaticoduodenectomy Biliary Tract Infection (Post-PD BTI)

In this part of the analysis, the factors, including age, sex, diagnosis, ASA-PS, relevant laboratory data, and information on PBI, were analyzed. The result of this part of the analysis demonstrates whether the PBI or a specific type of PBI is a risk factor for post-PD BTI.

### 2.4. Comparison of the Clinical Outcome between Patients with Different Types of PBI: ERCP vs. PTCD

In this part of the analysis, we further selected patients undergoing PBI and categorized them into the ERCP and PTCD groups. The interest outcomes included post-PD BTI and major surgical complications (Clavien–Dindo Classification ≥ 3).

### 2.5. Microbiological Study of Patients Who Underwent PBI and with a Positive Yield of Intraoperative Bile Culture

The results of bile culture with positive yield from either the ERCP or PTCD group were analyzed. The average number and species of intraoperative bile culture pathogens will be demonstrated. The difference in growth patterns between the two groups was also illustrated using a graphical method.

### 2.6. Statistical Analysis

Continuous variables are presented as means and standard deviations, while categorical variables are presented as numbers and percentages. We used Python (version 3.10.4), an open-source statistical software with the appropriate statistical packages, for analysis. Continuous variables were compared using Student’s *t*-test, while the χ^2^ test was used to analyze nominal data. Fisher’s exact test was applied if the test was not feasible due to statistical assumptions. A value of *p* < 0.05 was considered statistically significant.

## 3. Results

A total of 689 patients underwent PD during the nine-year study period. After excluding patients without intraoperative bile culture for later analysis for outcome and microbiology (n = 397) and ERCP performed at other institutes (n = 13), 279 patients were enrolled as eligible subjects. The flowchart of patient selection is expressed in Figure 1.

### 3.1. Risk Factor of Post-Pancreaticoduodenectomy Biliary Tract Infection (Post-PD BTI)

In the analysis of the risk factors of post-PD BTI, 279 patients were included, of which 27 patients (9.7%, 27/279) were diagnosed with post-PD BTI. In the present work, only a borderline p-value was observed in the ERCP procedure (*p* = 0.05, regarding ERCP biopsy) concerning higher rates in post-PD BTI, while the demographic and laboratory parameters, including most preoperative biliary interventions (pre-op PTCD, pre-op ERCP, ERCP stent, EPT), the yield of intraoperative bile culture, and preoperative BTI, were not significant (Table 1).

### 3.2. Impact of Preoperative Biliary Intervention on Post-PD BTI

In our analysis of the comparison of ERCP and PTCD, 164 patients fulfilled our selection criteria (Figure 1), of which 76.2% (n = 125) had ERCP and 23.8% (n = 39) had PTCD. The positive yield of intraoperative biliary culture was significantly higher in patients who underwent ERCP than in PTCD (90.4% vs. 41.0%, *p* < 0.001). Though insignificant, major postoperative complications occurred more frequently in the ERCP group (n = 12, 9.6%) than in the PTCD group (n = 0, 0%), as shown by the Clavien–Dindo Score. Over and above that, although not statistically significant, a trend of higher post-PD BTI (13.8% vs. 2.7%) and BTI-related septic shock (n = 5 vs. 0, 4.0% vs. 0%) was noticed in the ERCP group. The overall mortality rate was 3.2%, and all mortality occurred in the group undergoing ERCP. Clinical information and postoperative outcomes of patients who underwent preoperative biliary intervention are stated in Table 2 and Table 3.

### 3.3. Microbiological Analysis: ERCP vs. PTCD

The patterns of microbes’ growth differed between the ERCP and the PTCD groups. The average number of species was significantly higher (*p* = 0.046) for the ERCP group (two per culture) than for the PTCD group (one microbe per culture). *Acinetobacter* spp. in the bile was significantly higher in the PTCD group (*p* = 0.002). Conversely, *Enterococcus* spp. (*p* = 0.005) and yeast-like species (*p* = 0.01) were found to be significantly higher in the ERCP group. In addition, yeast-like species accounted for 14.77% of all microbes grown in the patients who underwent ERCP, with none seen in the patients who underwent PTCD (Figure 2). The growth pattern of pathogens in bile culture from patients of ERCP and PTCD groups is expressed in Figure 3.

## 4. Discussion

While we have mentioned that infection complication was the most reported complication regarding PBI, the present study focused only on one of these infection complications, post-PD BTI. The result showed that the ERCP procedure appeared to have a trend of increasing post-PD BTI risk with borderline statistical significance. Since only patients with intraoperative bile culture were enrolled, we demonstrated that ERCP increased bile culture’s positive yield. Further analysis of bile culture results illustrated the spectrum of gastrointestinal flora that dominated in the bile culture of patients under preoperative ERCP. In addition, the number of microbes from ERCP patients was also higher than that from the PTCD group.

In the present study, post-PD BTI occurred in 9.7%, based on the diagnostic criteria mentioned in Section 2. Currently reported incidence in previous studies ranged from 6% to 11% [20,21,22,23]. Therefore, the incidence of post-PD BTI was not much different from previous studies. While PBI had been mentioned as a risk factor for post-PD complications, especially infection complications, the current study did not relate PBI to post-PD BTI [22,23]. In our study, we did not identify PBI as a risk factor for post-PD BTI. While further dissecting the impact of different types of PBI on post-PD BTI, we did not identify relevant studies.

Our work was the first to compare this influence from different types of PBI on post-PD BTI, while scarce works focused on the influences of various kinds of PBI on post-PD complications. A Korean study with a cohort of 849 patients concluded that endoscopic biliary drainage was inferior to percutaneous biliary drainage regarding postoperative complications [24]. Another study from China claimed that endoscopic biliary drainage with a stent increased postoperative complications [17]. In the present work, we mainly focused on one specific complication: post-PD BTI. According to our result, rates of post-PD BTI (ERCP vs. PTCD, 13.8% vs. 2.7%) and related septic shock (ERCP vs. PTCD, 4.0% vs. 0%) were higher in the ERCP group numerically, although no statistical significance was noted. The only statistically significant adverse effect was the positive yield rate of intraoperative bile culture.

Increased positive yield of intraoperative bile culture has been reported in previous studies, and this condition did impose risk on postoperative complications [25,26,27,28,29]. In our series of comparisons of preoperative biliary interventions regarding ERCP and PTCD, this study’s results showed a significantly higher positive yield of intraoperative biliary culture in patients who underwent ERCP. In addition to positive yield, polyclonal yield was also noted in patients undergoing biliary stent placement via ERCP [27,30], and the present study also demonstrated this phenomenon. Since we mainly focused on post-PD BTI, we did investigate whether the positive yield of intraoperative bile culture influences the outcome (not illustrated in the current article), and the result demonstrated no significance. Current evidence from the literature and our work can determine two facts: (1) PBI, especially the endoscopic method with stent placement [27,30,31], did increase the positive yield of intraoperative bile culture, and (2) positive bile culture was relevant to increase surgical morbidity, although not specific to post-PD BTI.

In the present work, the spectrum of microbes in the intraoperative bile culture of the ERCP group was different from that of the PTCD group. In general, gastrointestinal microbes are the dominant flora identified from intraoperative bile cultures, and this phenomenon is more evident in the group of patients with ERCP and stent placement [27,29,30,32]. In our study, the ERCP group had more pathogens belonging to the genre of gastrointestinal microbes, and this may coincide with most patients of ERCP groups undergoing EPT and stent placement. On the contrary, more airborne and healthcare-related pathogens were identified in the PTCD group. This was reasonable since most of these patients would stay at the hospital for days or weeks before PD. Another noteworthy point is that a large portion of bile culture yield from patients undergoing ERCP had fungal pathogens identified. However, the impact of this phenomenon cannot be elucidated in the present work.

Several limitations need to be further discussed and investigated in the present study. Firstly, considering our study’s retrospective methodology and the fact that we excluded cases without bile culture taken, in such a study base the post-PD BTI incidence and further analysis concerning it could only be estimated. Since the present study was not prospective work, intraoperative culture was taken more often when suspected infection was based on perioperative findings, and this may overestimate the actual incidence in our institute. Therefore, we believe the actual incidence should be lower than 9.7% in our institute. Secondly, due to complicated conditions after surgery, the diagnosis of post-PD BTI is occasionally vague and challenging. Thirdly, we did not observe a long follow-up duration of post-PD BTI in our study, and only early post-PD BTI occurring at the same hospitalization was identified due to other possible risk factors of late post-PD BTI, such as bilio-enteric stricture, which may have disrupted our focus. So, we could not determine if preoperative biliary intervention increases the risk of late post-PD BTI. Fourthly, we did not consider the difficulty in biliary cannulation in ERCP or PTCD as a factor affecting the risk of post-PD BTI [33]. The condition of difficult cannulation could not be precisely identified in the records by retrospective review. Therefore, our study did not determine whether difficulty in biliary cannulation could potentially increase the risk of post-PD BTI. Another shortcoming was that operations were performed by different surgeons in our institute. Although the surgeons used similar surgical techniques, confounding factors related to various surgeons due to non-uniformity may still have an impact.

## 5. Conclusions

In conclusion, a clear benefit of PBI remains controversial for patients with a plan of conducting PD, since preoperative pathologic diagnosis or biliary drainage is not always necessary and may increase postoperative surgical morbidity.

## Figures and Tables

**Figure 1 jcm-13-04150-f001:**
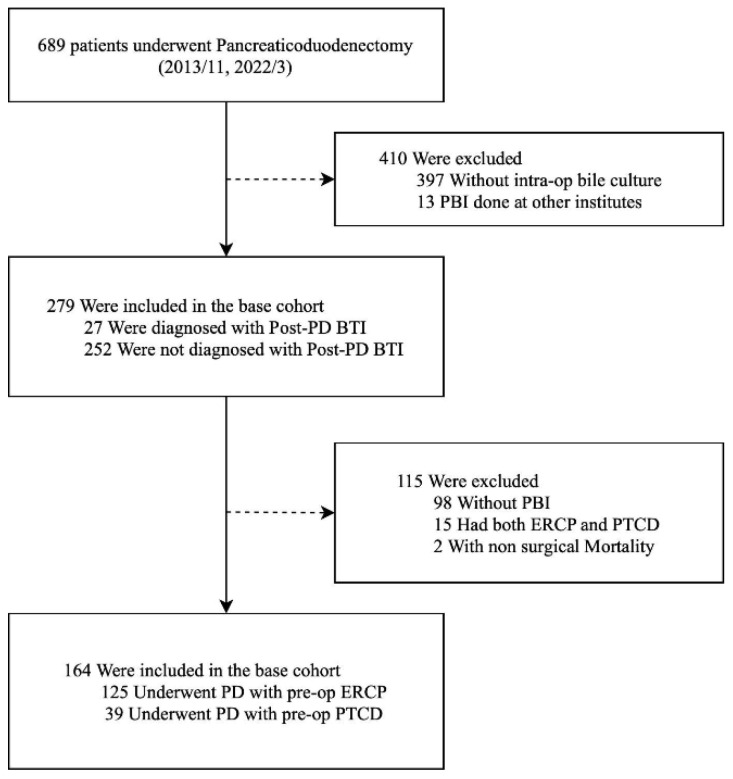
Flowchart of patient stratification; BTI, biliary tract infection; ERCP, endoscopic retrograde cholangiopancreatography; op, operative; PBI, preoperative biliary intervention; PD, pancreaticoduodenectomy; PTCD, percutaneous transhepatic cholangiography and drainage.

**Figure 2 jcm-13-04150-f002:**
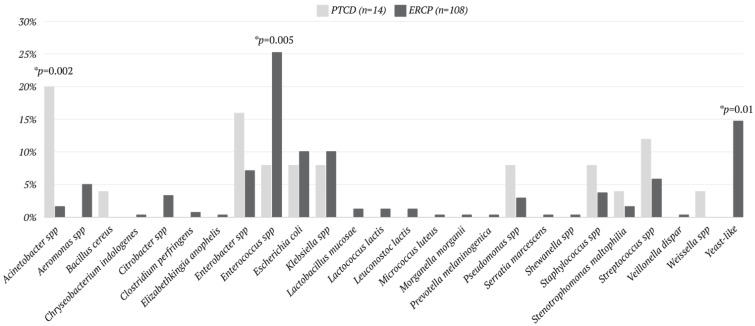
Biliary pathogens identified with a positive yield of intra-operative bile culture in patients undergoing pancreaticoduodenectomy; ERCP, endoscopic retrograde cholangiopancreatography; PBI, preoperative biliary intervention; PD, pancreaticoduodenectomy; PTCD, percutaneous transhepatic cholangiography and drainage; spp, species. * statistically significant.

**Figure 3 jcm-13-04150-f003:**
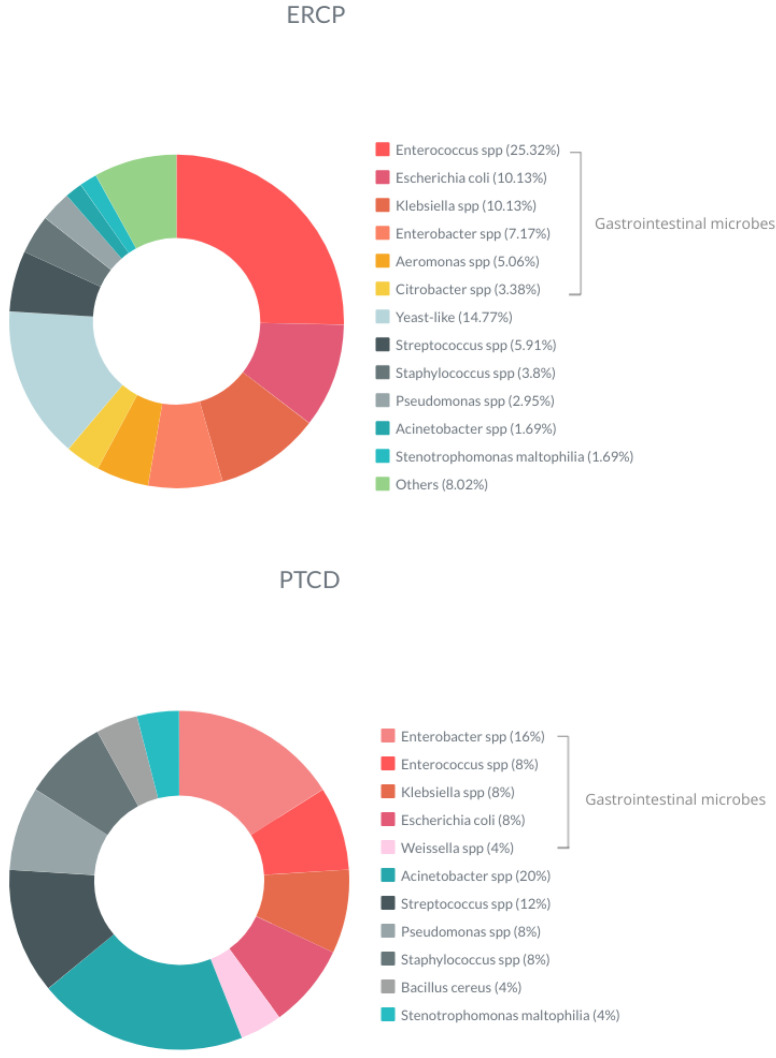
The growth pattern of pathogens in bile culture from patients of ERCP and PTCD groups; ERCP, endoscopic retrograde cholangiopancreatography; PTCD, percutaneous transhepatic cholangiography and drainage; spp, species.

**Table 1 jcm-13-04150-t001:** Risk analysis of post-pancreaticoduodenectomy biliary tract infection.

Study Population, n = 279				
		Post-op BTI (−)	Post-op BTI (+)	*p* Value
		252	27	
Age, mean (SD)		65.0 (11.5)	64.8 (14.5)	0.938
Sex (male), n (%)		149 (59.1)	19 (70.4)	0.354
Indication, n (%)	Pancreatic CA	119 (47.2)	12 (44.4)	0.777
	Ampullary CA	64 (25.4)	6 (22.2)	
	Biliary CA	41 (16.3)	7 (25.9)	
	Gastroduodenal CA	13 (5.2)	1 (3.7)	
	Others	15 (6.0)	1 (3.7)	
ASA-PS ≥ 3, n (%)	yes	116 (46.0)	14 (51.9)	0.709
WBC, mean (SD)		7.6 (2.9)	8.2 (3.1)	0.387
Hb, mean (SD)		11.7 (1.9)	11.6 (3.0)	0.823
PLT, mean (SD)		283.1 (97.0)	291.5 (109.3)	0.704
T. Bil, mean (SD)		3.5 (4.6)	2.7 (2.2)	0.150
AST, mean (SD)		72.9 (85.8)	66.0 (56.7)	0.580
ALT, mean (SD)		95.6 (105.3)	81.5 (77.1)	0.406
ALP, mean (SD)		239.8 (192.4)	202.8 (142.9)	0.243
Albumin, mean (SD)		4.5 (11.2)	3.6 (0.6)	0.295
CA199, mean (SD)		935.8 (3778.1)	916.6 (3528.0)	0.980
CEA, mean (SD)		13.3 (92.3)	3.1 (2.8)	0.110
ERCP status, n (%)	yes	126 (50.0)	18 (66.7)	0.149
Pre-op PTCD, n (%)	yes	50 (19.8)	3 (11.1)	0.400
ERCP stent, n (%)	yes	112 (49.1)	17 (68.0)	0.114
EPT, n (%)	yes	113 (49.6)	17 (68.0)	0.123
ERCP Biopsy, n (%)	yes	112 (49.1)	18 (72.0)	0.05
Pre-op BTI, n (%)	yes	70 (27.8)	12 (44.4)	0.113
HJ stent, n (%)	yes	120 (47.6)	13 (48.1)	1.000
Bile culture, n (%)	Positive	152 (60.3)	18 (66.7)	0.663

* Statistically significant. Abbreviations: ASA-PS, American Society of Anesthesiologists Performance Score; AST, aspartate aminotransferase; ALT, alanine aminotransferase; ALP, alkaline phosphatase; BTI, biliary tract infection; CA, cancer; CA199, carbohydrate antigen 19-9; CEA, carcinoembryonic antigen; EPT, endoscopic papillotomy; ERCP, endoscopic retrograde cholangiopancreatography; Hb, hemoglobin; HJ stent, hepaticojejunostomy with stent; PTCD, percutaneous transhepatic cholangiography and drainage; PLT, platelet; T.Bil, total bilirubin; WBC, white blood cell.

**Table 2 jcm-13-04150-t002:** Stratification of preoperative biliary intervention.

Study Population, n = 164				
		ERCP (+)	PTCD (+)	*p* Value
		125(76.2)	39(23.8)	
Age, mean (SD)		65.4 (11.4)	67.5 (10.6)	0.289
Sex(male), n (%)		82 (65.6)	24 (61.5)	0.786
Indication, n (%)	Pancreatic CA	36 (28.8)	28 (71.8)	<0.001 *
	Ampullary CA	49 (39.2)	5 (12.8)	
	Biliary CA	34 (27.2)	3 (7.7)	
	Gastroduodenal CA Others	1 (0.8) 5 (4.0)	3 (7.7)	
ASA-PS ≥ 3, n (%)	Yes	67 (53.6)	14 (35.9)	0.081
WBC, mean (SD)		7.6 (2.7)	7.7 (3.0)	0.938
Hb, mean (SD)		11.7 (2.1)	11.3 (1.8)	0.282
PLT, mean (SD)		283.7 (93.8)	286.2 (119.2)	0.918
T. Bil, mean (SD)		2.8 (4.0)	4.8 (5.5)	0.084
AST, mean (SD)		68.5 (78.4)	54.9 (31.7)	0.149
ALT, mean (SD)		80.0 (70.8)	94.9 (78.6)	0.370
ALP, mean (SD)		225.6 (173.7)	222.5 (131.1)	0.923
Albumin, mean (SD)		5.0 (14.6)	3.6 (0.4)	0.322
CA199, mean (SD)		734.4 (3773.5)	2488.8 (6333.4)	0.176
CEA, mean (SD)		3.8 (9.1)	5.9 (11.0)	0.383
Pre-op BTI, n (%)	yes	51 (40.8)	10 (25.6)	0.128

* Statistically significant. Abbreviations: ASA-PS, American Society of Anesthesiologists Performance Score; AST, aspartate aminotransferase; ALT, alanine aminotransferase; ALP, alkaline phosphatase; BTI, biliary tract infection; CA, cancer; CA199, carbohydrate antigen 19-9, CEA, carcinoembryonic antigen; ERCP, endoscopic retrograde cholangiopancreatography; Hb, hemoglobin; op, operative; PTCD, percutaneous transhepatic cholangiography and drainage; PLT, platelet; T. Bil, total bilirubin; WBC, white blood cell.

**Table 3 jcm-13-04150-t003:** Outcome comparison of preoperative biliary intervention.

Study Population, n = 164					
		ERCP(+)	PTCD(+)	*p* Value	
		125 (76.2)	39 (23.8)	Univariable analysis	Multivariable analysis
Bile culture, n (%)	Positive	113 (90.4)	16 (41.0)	<0.001 *	0.002 *
Clavien–Dindo classification ≥ 3, n (%)	yes	12 (9.6)		0.071	0.999
Post-op BTI, n (%)	yes	17 (13.8)	1 (2.7)	0.076	0.999
BTI-related septic shock, n (%)	yes	5 (4.0)		0.340	NA
Mortality, n (%)	yes	4 (3.2)		0.573	NA

* Statistically significant. Abbreviations: BTI, biliary tract infection; ERCP, endoscopic retrograde cholangiopancreatography; op, operative; PTCD, percutaneous transhepatic cholangiography and drainage; NA, not applicable.

## Data Availability

The data presented in this study are available on request from the corresponding author.

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
