# Peer review of "The Clinical Impact of Different Types of Preoperative Biliary Intervention on Postoperative Biliary Tract Infection of Patients Undergoing Pancreaticoduodenectomy"

_jcm, 2024, doi:10.3390/jcm13144150_

Round 1
Reviewer 1 Report
Comments and Suggestions for Authors
The topic is interesting and new but some aspects should be improved.
The methods section lacks of several technical details on the procedures, for example how many patients had difficult biliary cannulation during ERCP? How was it managed? in this regard, the authors should cite a relevant reference (PMID: 34543649 )
The authors should comment on the technical details of PTBD too.
Was there a cross-over between the two treatments in case of failure?
Why the authors focused only on the risk of infections and not also on other adverse events such as pancreatitis? If so, the authors should specify in the title and abstract that only infectious AEs were considered
Author Response
Q1. The topic is interesting and new but some aspects should be improved. The methods section lacks of several technical details on the procedures, for example how many patients had difficult biliary cannulation during ERCP? How was it managed? in this regard, the authors should cite a relevant reference (PMID: 34543649 )
ANS: Thank you for your precious comments. Indeed, the difficult cannulation of ERCP dose imposes a risk of procedure-related complications. Since the retrospective approach of our manuscript, we can only gain information on individual ERCP by reviewing the reports and the difficult cannulation cannot be identified in these reports. Nonetheless, the cohort of the first part of the analysis (n = 279) included 4 patients with occurrences of unsuccessful ERCP procedures. While 3 of these 4 patients underwent repeated ERCP, the rest of 1 underwent PTCD. Notably, patients subjected to both ERCP and PTCD procedures were excluded when conducting the second part of the analysis, the comparison between ERCP vs. PTCD (n = 164). In the second part of the analysis, 15 cases who underwent both ERCP and PTCD were excluded and 1 of the 15 patients was also the 1 of the 4 patients suffering from failure ERCP at the first attempt of the ERCP procedure.
In the methodology “Material and Methods: 2.2 Preoperative Biliary Interventions (PBI) and Surgical Procedures”, we have included additional details regarding ERCP failure. We acknowledged the potential impact of difficult biliary cannulation during ERCP on procedure duration and, consequently, on the infection rate. This factor has been incorporated into the limitations section of our discussion, with relevant references cited.
We have also provided technical descriptions of both ERCP and PTCD procedures in the "Material and Methods: 2.2 Preoperative Biliary Interventions (PBI) and Surgical Procedures" section. These additions aim to enhance the comprehensiveness and clarity of our study methodology.
Q2. The authors should comment on the technical details of PTBD too.
ANS: Thank you for the suggestion. We have added technical details of ERCP and PTCD in the section: “Material and methods: 2.2 Preoperative biliary interventions (PBI) and surgical procedures”.
Q3. Was there a cross-over between the two treatments in case of failure?
ANS: Thank you for the inquiry. If the initial procedure of ERCP failed or was insufficient, repeated ERCP or PTCD would be arranged. The decision for repeated ERCP or PTCD was made by clinicians based on patients’ condition, such as the bile duct condition in tomography images (CT or MRI), and clinicians’ preference. There were 15 cases who both underwent ERCP and PTCD, and they were excluded in the comparative analysis investigating the efficacy of ERCP versus PTCD methodologies.
Q4. Why the authors focused only on the risk of infections and not also on other adverse events such as pancreatitis? If so, the authors should specify in the title and abstract that only infectious AEs were considered.
ANS: In this retrospective study, 279 cases of pancreaticoduodenectomy were conducted at our institute after PBI. While 12 cases manifested post-PBI pancreatitis, no instances of post-pancreaticoduodenectomy pancreatitis were observed. As outlined in the “Introduction” section, paragraphs 2 and 3, our institutional experience underscores the impact of severe or fulminant postoperative biliary infections on patient outcomes and survival. This clinical observation prompted our investigation into whether scientific evidence aligns with our experience, so we just focused on post-PD biliary infection. Through this focused investigation, we aim to elucidate the clinical implications thereof.
Reviewer 2 Report
Comments and Suggestions for Authors
Conspicuous case history.
Topic debated in literature.
Some aspects should be better specified by the authors.
Do all patients undergo preoperative biliary drainage?
Do patients undergo drainage with any level of hyperbilirubinemia?
How many days after bile drainage is surgical treatment performed?
How many patients have undergone neoadjuvant treatments?
Have any differences been noted between patients with cephalopancreatic carcinoma and patients with carcinoma in other periampullary locations?
Author Response
Q1. Conspicuous case history. Topic debated in literature. Some aspects should be better specified by the authors. Do all patients undergo preoperative biliary drainage?
ANS: Thank you for your inquiry. In this retrospective study, spanning 689 cases of pancreaticoduodenectomy conducted at our institute, 416 cases underwent surgery directly without preoperative biliary drainage. The inclusion criteria for our study encompassed patients who had undergone preoperative biliary drainage before undergoing therapeutic pancreaticoduodenectomy for malignancies or suspected malignancies affecting the pancreatic or periampullary regions. Conversely, patients who had not undergone preoperative biliary drainage preceding pancreaticoduodenectomy were excluded from the study at the beginning.
Q2. Do patients undergo drainage with any level of hyperbilirubinemia?
ANS: Thank you for your inquiry. Patients who underwent biliary drainage due to obstructive jaundice typically presented with a certain level of hyperbilirubinemia, ranging from a median total bilirubin level of 10-14 mg/dL, and this is the consensus of the MDT (multidisciplinary team) in our institute. However, ERCP may be arranged under some consideration, such as the presentation of biliary tract infection and the possibility of diagnosis, for which surgery may not be the first consideration. In the past 1 to 2 years, patients with suspicious pancreatic cancer (resectable disease) may have ERCP due to the consideration of neoadjuvant chemotherapy in our institute.
Q3. How many days after bile drainage is surgical treatment performed?
ANS: Thank you for your inquiry. The timing of surgical intervention was predicated upon individual clinical assessments conducted by the surgeon. Typically, this intervention occurred within a timeframe ranging from 7 to 21 days following biliary drainage in our cohort. The variance in timing was contingent upon several factors. Firstly, if complications arise from biliary drainages, such as pancreatitis, hemorrhaging, or infection, priority was given to stabilizing the patient before surgery. Secondly, in cases where patients underwent ERCP biopsy concurrent with drainage, surgery was not arranged until the biopsy results were issued. The details of the timing of the surgery were discussed in the section: “Material and methods: 2.2 Preoperative biliary interventions (PBI) and surgical procedures”.
Q4. How many patients have undergone neoadjuvant treatments?
ANS: Thank you for your inquiry. None of the patients in our study population underwent neoadjuvant chemoradiotherapy. The data for our research spanned from June 2013 to March 2022. Neoadjuvant treatment and conversion surgery were not implemented in our institute until 2022, and the population eligible for such treatments was notably small. Those who did receive neoadjuvant chemoradiotherapy in our institute typically had borderline resectable pancreatic cancer or clinical lymph node-positive patients.
Q5. Have any differences been noted between patients with cephalopancreatic carcinoma and patients with carcinoma in other periampullary locations?
ANS: In the univariate analysis, females exhibited a higher prevalence in pancreatic cancer cases compared to males (46.6% vs. 33.8%, P=0.04). Additionally, levels of preoperative total bilirubin and ALT were elevated in patients diagnosed with cephalopancreatic cancer, with mean levels of 4.4 mg/dL and 122.9 U/L, respectively. In contrast, patients with periampullary cancer demonstrated lower mean total bilirubin levels of 2.6 mg/dL and ALT levels of 71.3 U/L, with corresponding P values of 0.002 and <0.001, respectively. The rates of preoperative biliary tract infection and positive intraoperative bile culture were significantly higher in patients with other cancers compared to pancreatic cancer.
Upon multivariate analysis, no significant differences were observed in patient stratification or other outcome comparisons regarding postoperative biliary tract infection, BTI-related septic shock, and mortality. However, the positive yield of intraoperative biliary culture was higher in patients with carcinoma located in other periampullary regions than those with pancreatic cancer (73.6% vs. 43.6%, P= 0.009). Also, the rate of preoperative biliary tract infection was also higher in the non-pancreatic cancer group (35.1% vs. 22.9, P=0.023)
While our study primarily focused on assessing the impact of preoperative biliary drainage, the observed variations among different types of cancer are noteworthy. Following is the table comparing pancreatic cancer and other cancers.
|
comparison between pancreatic CA and other Cas |
||||||
|
|
Overall |
Pancreatic CA |
others |
Univariate analysis |
Multivariate analysis |
|
|
n |
|
279 |
131 |
148 |
||
|
Age, mean (SD) |
|
65.0 (11.5) |
65.3 (10.8) |
64.8 (12.1) |
0.717 |
NA |
|
Sex, n (%) |
Female |
111 (39.8) |
61 (46.6) |
50 (33.8) |
0.040* |
0.64 |
|
|
Male |
168 (60.2) |
70 (53.4) |
98 (66.2) |
NA |
|
|
ASA≥3, n (%) |
yes |
130 (46.6) |
64 (48.9) |
66 (44.6) |
0.554 |
NA |
|
WBC, mean (SD) |
|
7.7 (2.9) |
7.5 (3.1) |
7.8 (2.8) |
0.328 |
NA |
|
Hb, mean (SD) |
|
11.7 (2.0) |
11.8 (2.1) |
11.6 (2.0) |
0.448 |
NA |
|
PLT, mean (SD) |
|
284.0 (98.2) |
289.0 (100.1) |
280.1 (96.9) |
0.485 |
NA |
|
TB, mean (SD) |
|
3.4 (4.5) |
4.4 (5.3) |
2.6 (3.5) |
0.002* |
NA |
|
AST, mean (SD) |
|
72.2 (83.0) |
86.5 (107.7) |
60.8 (54.1) |
0.026* |
NA |
|
ALT, mean (SD) |
|
94.1 (102.6) |
122.9 (131.3) |
71.3 (64.2) |
<0.001* |
NA |
|
ALKP, mean (SD) |
|
235.4 (187.4) |
251.8 (207.8) |
222.2 (168.8) |
0.255 |
NA |
|
Albumin, mean (SD) |
|
4.4 (10.6) |
3.7 (0.5) |
4.9 (14.1) |
0.356 |
0.180 |
|
CA199, mean (SD) |
|
933.8 (3745.5) |
1000.7 (3894.7) |
880.0 (3635.3) |
0.807 |
NA |
|
CEA, mean (SD) |
|
12.2 (87.1) |
23.8 (130.5) |
3.1 (5.7) |
0.108 |
NA |
|
Pre OP BTI, n (%) |
yes |
82 (29.4) |
30 (22.9) |
52 (35.1) |
0.035* |
0.023* |
|
Surgery type, n (%) |
PD |
191 (68.5) |
98 (74.8) |
93 (62.8) |
0.044 |
0.781 |
|
Bile culture, n (%) |
Positive |
170 (60.9) |
61 (46.6) |
109 (73.6) |
<0.001* |
0.009* |
|
Post op BTI, n (%) |
yes |
27 (9.7) |
12 (9.2) |
15 (10.1) |
0.943 |
NA |
|
complication≥3, n (%) |
yes |
23 (8.2) |
7 (5.3) |
16 (10.8) |
0.150 |
0.999 |
|
BTI-related septic shock, n (%) |
yes |
5 (1.8) |
2 (1.5) |
3 (2.0) |
1.000 |
NA |
|
Mortality, n (%) |
yes |
4 (1.4) |
1 (0.8) |
3 (2.0) |
0.625 |
NA |

Round 2
Reviewer 1 Report
Comments and Suggestions for Authors
The revised version of the manuscript is OK. Thank you!
Reviewer 2 Report
Comments and Suggestions for Authors
No further comments